# Immunological Response against Breast Lineage Cells Transfected with Human Papillomavirus (HPV)

**DOI:** 10.3390/v16050717

**Published:** 2024-04-30

**Authors:** Daffany Luana Santos, Bianca de França São Marcos, Georon Ferreira de Sousa, Leonardo Carvalho de Oliveira Cruz, Bárbara Rafaela da Silva Barros, Mariane Cajuba de Britto Lira Nogueira, Talita Helena de Araújo Oliveira, Anna Jessica Duarte Silva, Vanessa Emanuelle Pereira Santos, Cristiane Moutinho Lagos de Melo, Antonio Carlos de Freitas

**Affiliations:** 1Laboratory of Molecular Studies and Experimental Therapy, Department of Genetics, Federal University of Pernambuco, Av. Prof. Moraes Rego, 1235. Cidade Universitária, Recife 50670-901, Pernambuco, Brazil; daffany.luana@ufpe.br (D.L.S.); bianca.saomarcos@ufpe.br (B.d.F.S.M.); anna.jessica@ufpe.br (A.J.D.S.); vanessa.emanuelle@ufpe.br (V.E.P.S.); 2Keizo Asami Immunopathology Laboratory, Federal University of Pernambuco, Av. Prof. Moraes Rego, 1235. Cidade Universitária, Recife 50670-901, Pernambuco, Brazil; georon.sousa@gmail.com (G.F.d.S.); leonardo.oliveiracruz@ufpe.br (L.C.d.O.C.); barbara.sbarros@ufpe.br (B.R.d.S.B.); marianelira@gmail.com (M.C.d.B.L.N.); cristiane.moutinho@ufpe.br (C.M.L.d.M.); 3Department of Antibiotics, Federal University of Pernambuco, Recife 50670-901, Pernambuco, Brazil; 4Vitória Academic Center, Federal University of Pernambuco, Rua do Alto do Reservatório s/n, Bela Vista, Vitória de Santo Antão 55608-680, Pernambuco, Brazil; 5Patrick G Johnston center for Cancer Research, Queen’s University Belfast, University Road, Belfast BT7 1NN, UK; t.oliveira@qub.ac.uk

**Keywords:** oncogenes, MDA-MB-231, triple-negative, HPV

## Abstract

Breast cancer is the most common neoplasm worldwide. Viral infections are involved with carcinogenesis, especially those caused by oncogenic Human Papillomavirus (HPV) genotypes. Despite the detection of HPV in breast carcinomas, the virus’s activity against this type of cancer remains controversial. HPV infection promotes remodeling of the host’s immune response, resulting in an immunosuppressive profile. This study assessed the individual role of HPV oncogenes in the cell line MDA-MB-231 transfected with the E5, E6, and E7 oncogenes and co-cultured with peripheral blood mononuclear cells. Immunophenotyping was conducted to evaluate immune system modulation. There was an increase in CD4^+^ T cell numbers when compared with non-transfected and transfected MDA-MB-231, especially in the Treg profile. Pro-inflammatory intracellular cytokines, such as IFN-γ, TNF-α, and IL-17, were impaired by transfected cells, and a decrease in the cytolytic activity of the CD8^+^ and CD56^+^ lymphocytes was observed in the presence of HPV oncogenes, mainly with E6 and E7. The E6 and E7 oncogenes decrease monocyte expression, activating the expected M1 profile. In the monocytes found, a pro-inflammatory role was observed according to the cytokines released in the supernatant. In conclusion, the MDA-MB-231 cell lineage transfected with HPV oncogenes can downregulate the number and function of lymphocytes and monocytes.

## 1. Introduction

Breast cancer represents the most common neoplasm in the world, represented by a percentage of 11.7%, followed by lung cancer with 11.4%, and the most common in women, representing 25.4% of all types of cancer [1]. In 2020, 2.3 million women were diagnosed with breast cancer, responsible for 685,000 deaths worldwide [2]. According to data from INCA (National Cancer Institute—Brazil), in 2023, Brazil had an incidence of 73,610 cases of this type of cancer and 18,139 associated deaths [3]. There are two types of classification for breast cancer: histological (in situ or invasive ductal carcinoma and in situ or invasive lobular carcinoma) and molecular (Luminal A or B, super HER 2, and triple-negative) [4].

The relationship between hereditary genetic factors and the etiology of breast cancer has already been established. However, this cancer is also associated with several environmental and behavioral factors, such as prolonged exposure to estrogen due to early menarche, nulliparity, late menopause, exposure to high doses of ionizing radiation, regular consumption of alcohol, and a diet rich in fat [5].

The carcinogenesis process of several types of cancer, including cervical, anal, vulvar, vaginal, penile, and oropharyngeal, is associated with high-risk HPV persistent infections [6]. To date, nine types of HPVs (6, 11, 16, 18, 31, 33, 35, 45, and 52) have been identified in breast tumor samples from different populations around the world [5,7]. The variability in the percentage of virus detection and the controversial mechanism of infection are crucial points for establishing a causal association between HPV and mammary carcinogenesis [5,8,9,10,11]. There are three hypothesized pathways by which HPV would reach the breast tissue: (I) through oral sexual practices and microlesions in the skin of the breast or nipple; (II) through the blood in patients with cervical cancer; or (III) by metastasis, a possible secondary malignant transformation derived from a primary tumor in another organ [5].

Despite the established relationship between HPV and cervical cancer, the possible connection between this infectious agent and other types of cancer, such as breast cancer, still needs more research. In vitro studies using cellular transfection have been efficient in assessing the activity of HPV oncogenes against different types of cancers [12,13,14,15]. For example, a remodeling of the environment after HPV infection and the cells inserted in it provides a post-infection microenvironment (PIM) with immunosuppressive characteristics, leading to viral persistence and possible neoplastic development [16]. Recently, it has been observed that HPV infection may contribute to chemoresistance in breast cancer through genes involved in cellular apoptosis [17]. Immune cells and signaling molecules, such as members of the interleukin family, which are part of the tumor microenvironment, can affect cancer progression and development [18].

In this study, considering that HPV can alter the immune response and has already been detected in breast tumors, we investigated possible changes in breast adenocarcinoma cell lines transfected with HPV oncogenes and the role of immune cells in response to this target in vitro. Here, we assessed the individual influence of each HPV oncogene on the modulation of the immune response in breast carcinoma cells.

## 2. Materials and Methods

### 2.1. Cell Lineage

The MDA-MB-231 cell line (ATCC HTB-26) is a human breast cancer epithelial cell line. It is a cell derived from the metastasis of a mammary adenocarcinoma, triple negative. MDA-MB-231 has the following phenotype: CD44^+^CD24^−^/low. MDA-MB-231. The cells were cultured at 37 °C in Dulbecco’s Modified Eagle’s Medium (DMEM-Invitrogen^®^, Carlsbad, CA, USA) plus 10% fetal bovine serum (Gibco^®^, Waltham, MA, USA); 1% L-Glutamine (Sigma^®^, New York, NY, USA)—Complete DMEM—and passed through the priming process until reaching 70–80% confluence.

### 2.2. Isolation of Peripheral Blood Mononuclear Cells (PBMC)

About 8–10 mL of peripheral blood was collected in an EDTA tube from 5 healthy volunteer donors between 24 and 45 years, 2 men, and 3 women, which did not present recent (a month before the blood collection) infection (such as respiratory syndromes, hepatic or gastrointestinal compromise), metabolic syndromes or cancer. Peripheral blood mononuclear cells (PBMCs) were isolated from blood using a 1.077 g/mL Ficoll solution (300× *g*/30 min/26 °C) (GE Healthcare^®^, Chicago, IL, USA). After two washes with 1X PBS, the cells were resuspended in Dulbecco’s Modified Eagle’s Medium (DMEM-Invitrogen^®^) with 10% fetal bovine serum (Gibco^®^) added, 1% L-Glutamine (Sigma^®^) and antibiotics. Cells obtained from each volunteer were cultured in 75 cm^2^ culture flasks (24 h/5% CO_2_/37 °C). The next day, the suspended lymphocytes were centrifuged in PBS 1X (200× *g*/10 min) and counted using the Countess 3 cell counter (Thermo Fisher^®^, Waltham, MA, USA). From the same culture flask, the adhered macrophages were removed with the aid of 2% trypsin, followed by centrifugation with 1X PBS (200× *g*/10 min) and counted (Countess 3/Thermo Fisher^®^). Lymphocytes and macrophages were plated (48-well plates/10^5^ cells/well) in different cell culture schemes (see Section 2.4).

### 2.3. Production and Transfection of HPV Oncoproteins

The E5, E6, and E7 wild-type oncogene sequences (unaltered reference gene sequence) of HPV16, based on the sequence deposited in GenBank (K02718.1), were cloned using the expression vector for mammalian cells, pCDNA 3.1 (+)—Invitogen^®^. Clones were confirmed by restriction analysis and sequenced using the ABI PRISM BigDyeTM Terminator v3.1 ReadyReaction cycle sequencing kit (Applied Biosystems^®^, Waltham, MA, USA) in the ABI Prism 3100 automated DNA sequencer (Applied Biosystem^®^). After the cloning confirmation, the DNA of recombinant vectors was isolated by the Plus Maxi kit (Qiagen, Germantown, MD, USA) according to the manufacturer’s instructions.

### 2.4. Transfection of HPV Oncoproteins 

In 48-well cell culture plates, about 10^5^ tumor cells (MDA-MB-231) in each well were seeded in 1 mL of Dulbecco’s Modified Eagle’s Medium (DMEM-Invitogen^®^, Waltham, MA, USA) plus 10% fetal bovine serum (Gibco^®^); 1% L-Glutamine (Sigma^®^)—Complete DMEM. These tumor cells were transfected with 250 ng/uL of empty pcDNA 3.1 (+) vector, pcDNA-E5, pcDNA-E6, or pcDNA-E7 separately using Lipofectamine 3000 transfection reagent (Thermo Fisher) following the protocol recommended by the manufacturer. Cells were incubated at 37 °C in a 5% CO_2_ incubator for 24 h. Subsequently, RNA was extracted from cells of those pellets using the PureLink^TM^ RNA Mini Kit (Invitogen^®^) and used to confirm the efficiency of transfection of the E5, E6, and E7 by RT-qPCR. All reactions were performed in a LineGene9660 (Bioer, Miramar, FL, USA) thermos cycler, using SYBR green with GoTaq qPCR Master Mix (Promega, Madison, WI, USA) as a detection system and specific primers for E5 (Forwad: ACT GGC GTG CTT TTT GCT TTG and Reverse: GAC ACA GAC AAA AGC AGC GG), E6 (Forward: GAG AAA CTG CAA TGT TTC AGG ACC and Reverse: TGT ATA GTT GTT TGC AGC TCT GTG C) and E7 (Forwad: AGC TCA GAG GAG GAG GAT GA and Reverse: GAG AAC AGA TGG GGC ACA CA). To calculate the relative expression of the target genes, they were analyzed together with the reference genes ACTB (Forwad: AAGAGAGGCATCCTCACCCT and Reverse: TACATGGCTGGGGTGTTGAA) and GAPDH (Forwad: GAAGGTGGGGCTCATTTG and Reverse: TTAAAAGCAGCCCTGGTG). The 2^ΔΔct^ was calculated to analyze the relative expression of each oncogene in comparison with the endogenous genes (ACTB and GAPDH), as well as it was compared with the expression of the oncogenes in the C3 cell line (HPV16 positive uterine cervix cell) [19]. The Ct (threshold cycle) values greater than 35 or those that did not present an amplification value, were excluded from the following analysis steps.

### 2.5. In Vitro Stimulation of PBMC

When the transfected cells (as described in the previous section) reached 80% confluence, lymphocytes or macrophages were inserted into the cultures to investigate the immune stimulus. Lymphocytes and macrophages obtained, as described before (Section 2.2), were cultured separately in 48-well plates, previously seeded with tumor cells, using the cell culture schemes described in Figure 1. The cultured cells were maintained for 24 h under the stimulus of the tumor cells, and after this period, the cells were collected and marked for immunophenotyping, and the supernatants of the cultures were stored for cytokine investigation.

### 2.6. Immunophenotyping of PBMCs Stimulated In Vitro with Tumor Cells and Investigation of the Cytokines Produced in the Culture Supernatant

The immunophenotyping assay was conducted in vitro to investigate the immune response induced by tumor cells in PBMC through anti-CD3 (FITC or APC), -CD4 (FITC or APC), -CD8 (FITC or PE) surface antibodies, -CD56 (PEcy5.5 or APC), -CD25 (PercP), -CD14 (FITC), -B7.1 (PE), -B7.2 (APC), -HLA-DR (PercP). Intracellular antibodies anti-FoxP3 (PE), -perforin (FITC), granzyme (PercP), -IL-17 (PE), -IFN (APC), and IL-10 (PE) were also used. For specific identification of the MDA-MB 231 lineage, the anti-CD44 antibody was used (all antibodies were from BD^®^, Franklin Lakes, NJ, USA). The cytokines IL-2, IL-4, IL-6, IL-10, TNF-α, IFN-α, and IL-17 were investigated using the CBA kit (BD^®^) from the culture supernatant. All acquisitions were performed using flow cytometry (Accuri BD^®^, Ann Arbor, MI, USA), and 30,000 events in the P1 region (lymphocytes or monocytes gates) for cellular investigation and 2100 events for humoral investigation through cytokine production in experimental groups. Analyses were performed using the Accuri cytometry platform.

### 2.7. Statistical Analysis

Data distribution was verified using the Kolmogorov–Smirnov test. Only samples that followed a normal distribution were included in the analysis, and the Ordinary One-way ANOVA test was performed. A *p*-value < 0.05 was considered statistically significant. All statistical analyses were conducted using GraphPad Prism software version 9.0.0 (GraphPad Software, Inc., San Diego, CA, USA).

## 3. Results

### 3.1. Expression of Oncogenes E5, E6, and E7

To confirm the expression of viral oncogenes after transfection, an RTqPCR assay was performed using specific primers for oncogenes E5, E6, and E7. The presence of the three oncogenes introduced through transfection was identified in breast cells. It was evident that there were individual expressions of E5, E6, and E7 in the MDA-MB-231 line cell (Figure 2).

### 3.2. Stimulation Profile of CD4^+^ T Lymphocytes

For the investigation of the immunological stimulation profile caused by HPV in lymphocytes and monocytes, in vitro, breast tumor cells were transfected with the E5, E6, and E7 oncogenes and exposed to immune cells for 24 h (Figure 3A). Interesting results for regulatory lymphocytes (Treg) can be seen in Figure 3B, when transfected cells, especially E6 and E7, showed an increase in the suppressor profile of CD4^+^ lymphocytes (CD4^+^CD25^+^FOXP3^+^). The decrease in intracellular IFN-γ reinforces the regulatory profile found in the CD4^+^ lymphocytes, which were impaired by HPV oncogenes in transfected cells, as we can see in Figure 3C.

### 3.3. Cytotoxic Stimulation of CD8^+^ T Lymphocytes

The investigation of the cytotoxic action of CD8^+^ T lymphocytes against tumor cells showed no change in the count of these cells (Figure 4A). However, there was a statistical increase in the intracellular production of the perforin enzyme in relation to E6 and a tendency to E7 (Figure 4B); moreover, there was a decrease in the granzyme enzyme in E7 with statistical values (Figure 4B). Different results were found for Natural Killer T lymphocytes (CD56^+^) when cultured with MDA-MB 231 cells. There was a decline in these lymphocytes, especially in relation to the E6 and E7 genes (Figure 4C) followed by an increase in the cytotoxic response capacity due to the high perforin production and intracellular granzyme accumulation (Figure 4D). 

### 3.4. Intracellular Cytokine Profile of CD8^+^ and CD56^+^ T Lymphocytes

Similar to CD4^+^ T lymphocytes, intracellular cytokine production from cytotoxic cells was investigated. E6 and E7 oncogene-transfected cells promoted an IFN-γ cytokine decrease in both CD8^+^ and CD56^+^ lymphocyte subtypes (Figure 5A,D). The same occurred for IL-17 (Figure 5B,E) and IL-10 (Figure 5C,F), especially when there was E7 expression.

### 3.5. Stimulation Profile of CD14^+^ Monocytes

The immune response profile induced by monocytes against tumor cells showed a suppression of these immune cells promoted by E6 and E7 oncogenes (Figure 6A), followed by a decline in the co-stimulatory response of the B7.1 molecule (Figure 6B). There were no changes in B7.2 and HLA-DR co-stimulatory molecules.

### 3.6. Cytokines Released in the Supernatants of Experimental Groups

The study investigated the cytokines released in the supernatants of both lymphocyte and monocyte cultures. The findings from the lymphocyte and MDA-MB-231 cultures showed that transfected cells, particularly those with E6 and E7 HPV oncogenes, exhibited significantly elevated levels of IL-2, IL-17, IFN-γ, TNF-α, and IL-4 (Figure 7A–C,E,G). While IL-10 showed a differential increase in E5, the difference was not statistically significant (Figure 7F).

The results of monocytes and MDA-MB-231 cultures showed high production of TNF-α in cultures transfected with HPV oncogenes and increased IFN-y and IL-6, especially in the presence of E7 (Figure 8A–C). As observed in lymphocyte cultures, the cytokine IL-10 was differentially produced only in monocyte cultures with the empty vector and E5 transfected tumor cells (Figure 8D), and IL-6 was highly produced in all experimental groups, especially when E7 was transfected (Figure 8C). No changes were observed in the IL-2 and IL-17 cytokine profiles.

## 4. Discussion

Usually, the main tumor-infiltrating immune cells isolated from breast cancer are CD4^+^ and CD8^+^ T cells, in addition to B cells and tumor-associated macrophages. Otherwise, eosinophils, monocytes, and natural killer (NK) inactive cells are present in smaller amounts in the microenvironment of breast tumors. In general, CD4^+^ T helper 1 (Th1) cells, CD8^+^ cytotoxic T cells, NK cells, and M1 macrophages have a protective activity against tumor development, whereas Treg, CD4^+^ T cells (Th2), and M2 macrophages can promote tumor growth [20]. The viral oncoproteins E5, E6, and E7 are the most studied proteins and of utmost importance in the immune evasion process [21]. In HPV-infected tumors, there is a limitation in the expression levels of their genes (10–100 copies), aiming to reduce the presentation of viral antigens on MHC class I, thereby decreasing recognition by the innate immune system as well as diminishing the adaptive response [22]. 

In HPV-associated breast tumors, the viral genome is predominantly integrated into the host DNA and has a low copy number [5,23]. In cervical cancer, the integration of the HPV genome relies on the inactivation of the E2 gene, which acts as a transcriptional regulator of E6 and E7. The E2/E6 and E7 ratio is calculated to determine the virus’s physical status [23,24]. The integration status of HPV is also associated with the modulation of the immune signature. Genes specifically expressed in T cells (CD4^+^, regulatory, CD3^+^, and CD8^+^), as well as in NK cells and B cells, have been found to be significantly higher in head and neck tumors with active integration [25].

In addition, in most breast tumors, there is a low infiltration of T cells. However, in triple-negative breast tumors, a high number of tumor-infiltrating lymphocytes is observed [26]. However, our data showed a decrease in intracellular IFN-γ production caused by CD4^+^ T cells, consequently decreasing the production of Th1 cytokines. These findings are in line with a study by Meng Cao (2020), where a change in the profile of Th1 cytokines (IFN-y, IL-2, and TNF-α) to Th2 (IL-4) was observed and correlated with more severe HPV infections [27]. Another important result observed in CD4^+^ T lymphocytes, in the presence of E6 and E7 oncogenes, was the increase in FOXP3-positive cells. Treg cells are characterized by the expression of CD4^+^, CD25^+^, and Foxp3, since these cells are known to suppress antitumor immunity and block attacks, especially by cytotoxic lymphocytes, against malignant cells [28,29]. In fact, a study by Hayashi et al. (2022) in HPV-infected human oral squamous cell carcinoma samples demonstrated an increased number of Tregs in carcinomas suppressed antitumor immune responses, contributing to the onset and exacerbation of malignancies. Increased Treg can directly suppress CD8^+^ cell antitumor function [30]. 

The HPV oncoproteins caused a decrease in NK cell proliferation in oncogene-transfected cell cultures, and, similarly to the CD8^+^ lymphocyte subtype, there was a suppression of cytolytic response due to the low production of perforin and accumulation of intracellular granzyme. Perforin is integrated into the cell membrane, forming pores through which granzymes invade the target cell, causing cell death [31]. Thus, perforin plays a key role in delivering granzymes to target cells, promoting the cytotoxicity of T cells and NK cells. Decreasing perforin levels may indicate that antigen-specific immune cells are being eliminated [32]. A study in breast tumor patients showed a greater release of granzyme in the blood of patients compared to tissue samples, and the expression of granzyme and perforin in breast cancer cells was involved in cytotoxicity for these cells [33]. A previous study also pointed out that TNF-α is essential to perforin-mediated death in cervical keratinocyte cells expressing E7 oncoproteins and that IFN facilitated the death of these cells even in the absence of perforin [34]. 

In our study, we observed no change in the number of CD8^+^ T cells stimulated with breast tumor cells (with or without transfection). However, it is possible to notice a decrease in the production of granzyme in lymphocytes stimulated with tumorigenic cells transfected with E6 and E7. In addition, there was also a significant decrease in the intracellular production of IFN-γ and IL-17 in lymphocytes from E6 and E7 transfected CD8 T cells, decreasing the inflammatory profile expected from CD8^+^ T lymphocytes. CD8^+^ cells induce tumor clearance through the production of IFN-γ, TNF, and granzyme B [35]; however, the same was not found in our study, in which there was suppression of the activity of CD8^+^ T cells induced by the presence of oncoproteins E6 and E7, as well as an increase in Treg cells.

Similar to the CD8^+^ findings, there was a decrease in the intracellular production of IFN-γ and IL-17 in NK lymphocytes from E6 and E7 transfected cells. These findings corroborate studies carried out in cervical cancer, where the number of NK cells is suppressed in the presence of E6 and E7 of HPV16, which in turn has the ability to inhibit the synthesis of IFN-γ induced by IL-18 [36].

The macrophage profile was also analyzed. M1 phenotype macrophages characterize pro-inflammatory responses and are associated with a good prognosis for antitumor activities. M2 phenotype macrophages are associated with a poor prognosis, given that they promote tumor angiogenesis [37]. In our study, CD14^+^ cells decreased in the presence of E6 and E7 oncogenes, as well as their co-stimulatory molecules B7.1, not effectively stimulating the M1 profile. A lung cancer study detected B7.1 and B7.2 primarily in tumor-infiltrating macrophages, from which B7.1 was associated with a worse prognosis [38]. In cervical cancer cell lines, it is likely that the induction of M2 macrophages occurs, which may provide a tumor immunosuppressive microenvironment, thus acting positively on angiogenesis and metastasis processes [39]. Accordingly, two studies with breast cancer patients demonstrated a reduced expression of HLA-DR in CD14^+^ cells associated with mechanisms of immunosuppression [40,41]. According to Figueiredo (2019), the M2 macrophage profile negatively regulates HLA-DR and IL-12 and increases the expression of anti-inflammatory cytokines, such as IL-4 and IL-10 [42]. Our results regarding the cytokines released by macrophages demonstrate that only the E7 oncoprotein stimulated the release of IFN-γ. Cytokines IL-2, IL-17, and IL-4 were not stimulated significantly in the experimental groups, but TNF-α was stimulated by E5, E6, and E7 in the transfected cultures, and IL-6 was stimulated only by E7, demonstrating an increased M1 profile and decreased M2 due to the suppression of IL-10 release by monocytes in the presence of E6 and E7. These findings suggest a pro-inflammatory profile of the few monocytes found in the environment of HPV-associated in vitro cultured breast cancer cells. In agreement, when analyzing the lymphocyte supernatant cultures, E6 and E7 oncogenes mainly induced the release of Th1 (IL-2, IFN-y, and TNF-α) and Th17 (IL-17), leading to a pro-inflammatory profile too. 

In more advanced stages of ductal carcinoma, tumor tissues are infiltrated by abundant concentrations of T cells and M2 macrophages, which produce large amounts of IL-6, promoting metastasis of breast tumor cells. This was evidenced by the fact that IL-6 serum levels in patients with ductal carcinoma were significantly higher than in healthy women. As for the TNF-α levels of stage III carcinoma patients, they were also significantly higher in relation to healthy women. When comparing the levels of TNF-α in breast cancer patients (stage III) to those with stage I cancer and control patients (healthy), stage III levels were elevated. This study indicated that only patients with tumors in advanced stages secrete high levels of TNF-α [43]. In our analyses, we highlight the results of cells transfected with E6 and E7, which promoted the high release of cytokines, including IL-6 and TNF-α. Furthermore, IL-6 was stimulated in all experimental groups.

Our results highlight how breast tumor cells transfected with different HPV oncogenes can shape lymphocyte and monocyte activity in distinct ways. Despite being a limited study, given its use of an in vitro model involving a tumor cell line and a subset of immune cells isolated from PBMCs of various donors, the study revealed a robust and replicable pattern of alterations in lymphocyte and monocyte responses, regardless of potential variations among donors in immune cell response capacity. However, in vivo confirmation of these in vitro findings using samples from primary tumors is still needed.

## 5. Conclusions

This study presents new information about the HPV activity in breast cancer based on the evaluation of an in vitro cell co-culture model of the individual action of HPV16 oncogenes E5, E6, and E7 as the immune response of monocytes and lymphocytes. Overall, the results demonstrated that the main role of HPV in MDA-MB-231 was to modulate the immune response by increasing regulatory T cells and decreasing CD8^+^ and CD56^+^ T lymphocytes. Furthermore, the E6 and E7 oncogenes significantly reduce the expression of monocytes, activating even the M1 profile, unlike E5. This study highlights aspects of the HPV oncoproteins and breast line cells relationship. More studies are still needed to understand the mechanisms of HPV and its oncoproteins within the context of breast cancer and their repercussions for cancer prevention and treatment, including in vivo studies to validate the presented findings.

## Figures and Tables

**Figure 1 viruses-16-00717-f001:**
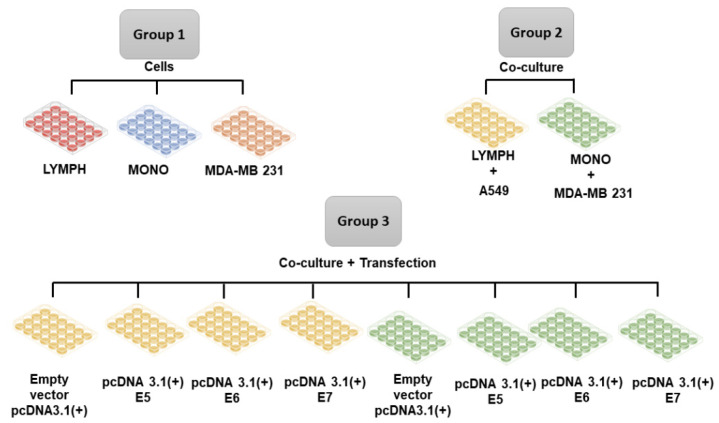
Group 1. Lymphocytes (LYMPH), monocytes (MONO), MDA-MB-231 cells; Group 2 MDA-MB-231 cells co-cultured with lymphocytes (LYMPH + MDA), MDA-MB-231 cells co-cultured with monocytes (MONO + MDA); Group 3 LYMPH + MDA cultured with pcDNA 3.1 (empty vector), LYMPH + MDA transfected with the E5 oncogene, LYMPH + MDA transfected with the E6 oncogene, LYMPH + MDA transfected with the E7 oncogene, MONO + MDA cultured with empty vector, MONO + MDA transfected with the E5 oncogene, MONO + MDA transfected with the E6 oncogene, MONO + MDA transfected with the E7 oncogene.

**Figure 2 viruses-16-00717-f002:**
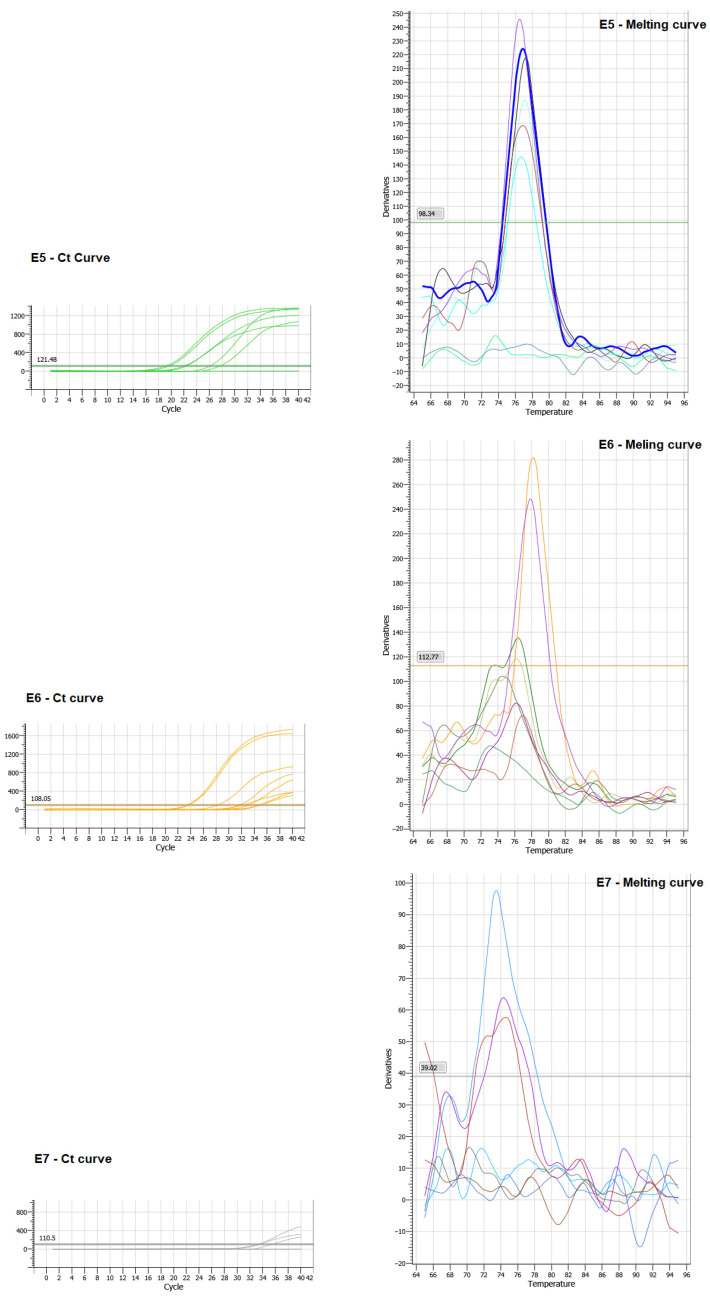
Ct and Melting curve referring to the expression of oncogenes E5, E6 and E7 transfected in MDA-MB-231 cells.

**Figure 3 viruses-16-00717-f003:**
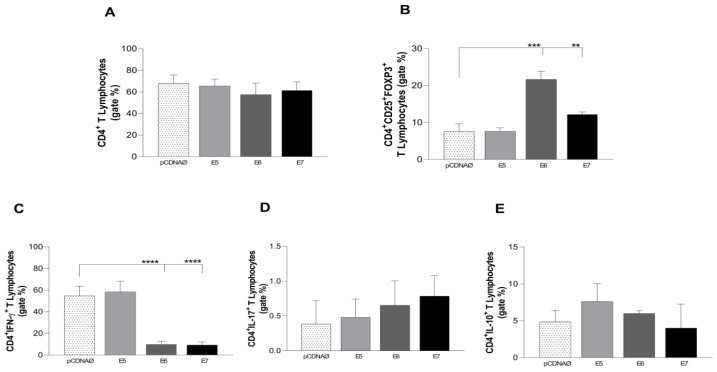
Stimulation profile of CD4^+^ T lymphocytes cultured with the MDA-MB 231 tumor cell line. The experimental groups were composed of the empty pcDNA vector and lymphocytes cultured with MDA-MB 231 transfected with the E5, E6, and E7 genes. (**A**)—Differential count of CD4^+^ T lymphocytes. Differential (**B**) count of the CD25^+^ subset of CD4^+^ T lymphocytes and its suppressor expression profile by the presence of the intracellular molecule FOXP3^+^. (**C**–**E**) The presence of IFN-γ, IL-17, and IL-10 intracellular cytokines in CD4^+^ lymphocytes, respectively. ** *p* < 0.01, *** *p* < 0.001, **** *p* < 0.0001. Error bars: standard error between samples.

**Figure 4 viruses-16-00717-f004:**
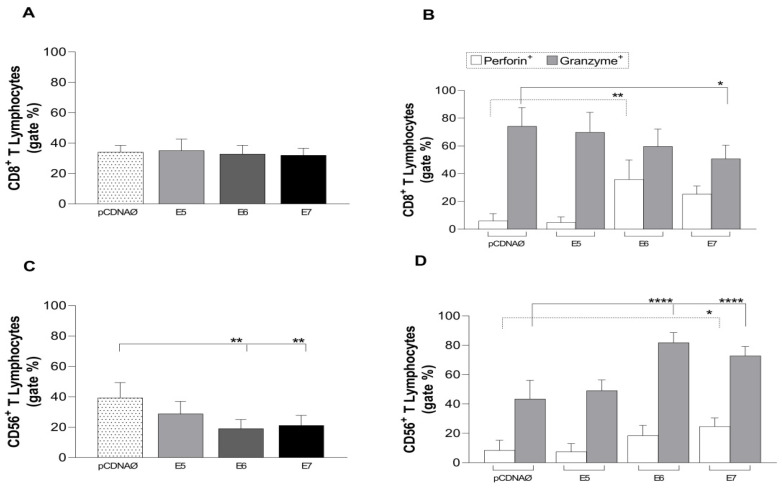
Stimulation profile of CD8^+^ and CD56^+^ T lymphocytes cultured with the MDA-MB 231 tumor line. Experimental groups: the empty pcDNA vector and lymphocytes cultured with transfected MDA-MB 231 with the E5, E6, and E7 genes. (**A**)—Differential count of CD8^+^ T lymphocytes. (**B**)—Intracellular production of perforin and granzyme in CD8^+^ T lymphocytes. (**C**)—CD56^+^ T lymphocyte differential count. (**D**)—Intracellular production of perforin and granzyme in CD56^+^ T lymphocytes. Asterisks represent statistical significance (* *p* < 0.05, ** *p* < 0.001, **** *p* < 0.0001). Error bars: standard error between samples.

**Figure 5 viruses-16-00717-f005:**
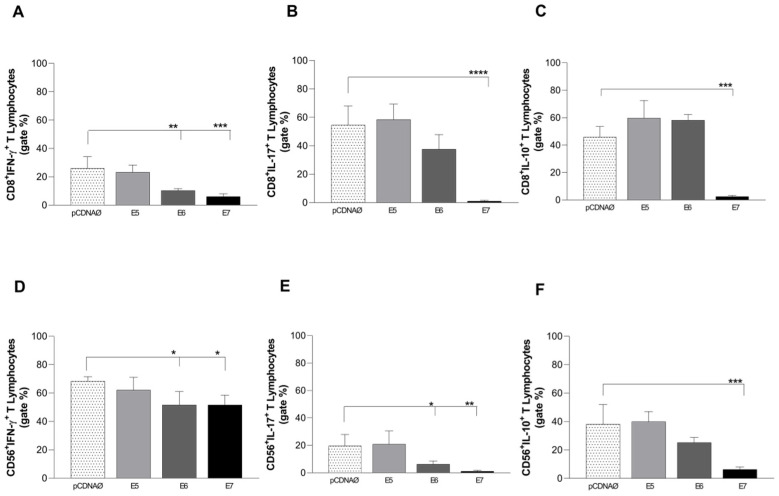
Intracellular cytokines are produced by CD8^+^ and CD56^+^ lymphocytes against tumor cells. (**A**–**C**)—CD8^+^ T lymphocytes producing IFN-γ, IL-17, and IL-10, respectively; (**D**–**F**)—CD56^+^ T lymphocytes producing IFN-γ, IL-17 and IL-10, respectively. * *p* < 0.05, ** *p* < 0.01, *** *p* < 0.001, **** *p* < 0.0001. Error bars: standard error between samples.

**Figure 6 viruses-16-00717-f006:**
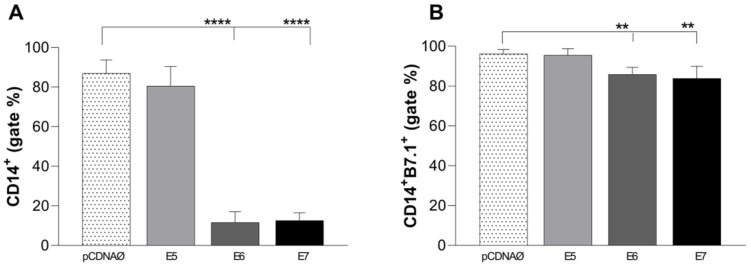
Stimulation profile of CD14^+^ monocytes cultured with MDA-MB-231 tumor cell lineage. Experimental groups: monocytes cultured with E5, E6, and E7 oncogenes and with the empty vector, transfected MDA-MB-231. (**A**)—Differential count of CD14^+^ monocytes. (**B**)—B7.1 surface costimulatory molecule production. ** *p* < 0.001, **** *p* < 0.0001. Error bars: standard error between samples.

**Figure 7 viruses-16-00717-f007:**
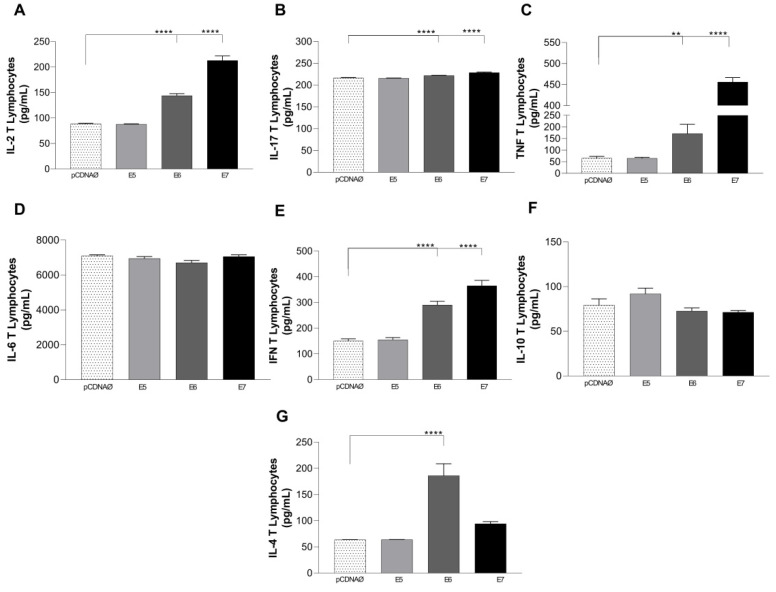
Cytokines are produced in the supernatants of lymphocyte and tumor cell cultures. (**A**–**G**)—cytokines IL-2, IFN-γ, TNF-α, IL-6, IL-17, IL-10, and IL-4, respectively. Asterisks represent statistical significance ** *p* < 0.001, **** *p* < 0.0001. Error bars: standard error between samples.

**Figure 8 viruses-16-00717-f008:**
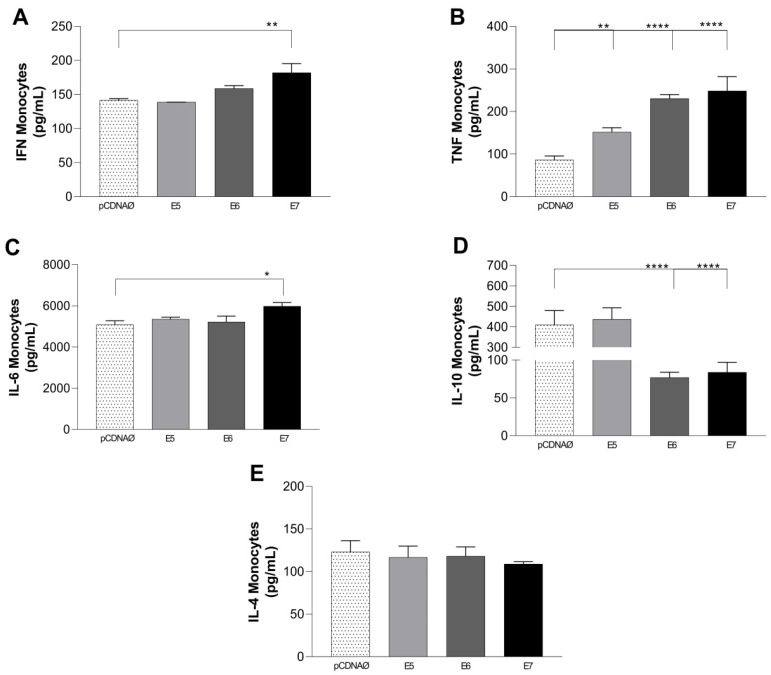
Cytokines are produced in the supernatants of monocyte and tumor cell cultures. (**A**–**E**)—IFN-y, TNF-α, IL-6, IL-10, and IL-4 cytokines, respectively. * *p* < 0.05, ** *p* < 0.01, **** *p* < 0.0001. Error bars: standard error between samples.

## Data Availability

The information provided in this study can be obtained by contacting the corresponding author upon request.

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
