# Peer review of "Immunological Response against Breast Lineage Cells Transfected with Human Papillomavirus (HPV)"

_viruses, 2024, doi:10.3390/v16050717_

Round 1
Reviewer 1 Report (Previous Reviewer 2)
Comments and Suggestions for Authors
The Authors corrected the first version of manuscript. Introduction and Discussion have been redrafted. I think that manuscript in current form may be accepted for publication.
Author Response
REPLY TO REFEREES
Recife, April 15, 2024
I am re-submitting the review article entitled “IMMUNOLOGICAL RESPONSE AGAINST BREAST LINEAGE CELLS TRANSFECTED WITH HUMAN PAPILLOMAVIRUS (HPV)” by Santos and São Marcos et al., for publication in Viruses.
All modifications are highlighted in the manuscript.
All the authors confirm that they saw and agreed to the submitted paper. The authors have been recognized as contributors and have agreed to their inclusion. The material is original, and it has been neither published elsewhere nor submitted for publication simultaneously. None of the authors has any potential financial conflict of interest related to this manuscript.
Review
1- The Authors corrected the first version of manuscript. Introduction and Discussion have been redrafted. I think that manuscript in current form may be accepted for publication.
Answer: Thank you for agreeing to the publication of the revised manuscript. We appreciate your support and look forward to the next steps in the publication process.
The complete contact information for corresponding author is: Dr. Antonio Carlos de Freitas (ORCID0000-0002-4957-9549), Av. Prof. Moraes Rego, 1235, Cidade Universitária, 50670-901, Recife-PE, Brazil. Fax: +55 81 21268512. E-mail: [email protected] .
Please do not hesitate to contact me if further information is needed.
Sincerely,
ANTONIO CARLOS DE FREITAS, PH.D
Associate Professor
Head of Laboratory of Molecular Studies and Experimental Therapy (LEMTE)
Department of Genetics
Federal University of Pernambuco
Recife, Pernambuco -Brazil

Reviewer 2 Report (Previous Reviewer 3)
Comments and Suggestions for Authors
The authors already submitted this manuscript and received the proper comments from the reviewers including mines. Unfortunately, zero effort was made by this team to improve the quality and soundness of the scientific side of their investigation and fix its substantial flaws. The authors should consider in the future rigorously respond to the comments they receive accordingly.
Comments on the Quality of English LanguageAverage
Author Response
Recife, April 15, 2024
I am re-submitting the review article entitled “IMMUNOLOGICAL RESPONSE AGAINST BREAST LINEAGE CELLS TRANSFECTED WITH HUMAN PAPILLOMAVIRUS (HPV)” by Santos and São Marcos et al., for publication in Viruses.
All modifications are highlighted in the manuscript.
All the authors confirm that they saw and agreed to the submitted paper. The authors have been recognized as contributors and have agreed to their inclusion. The material is original, and it has been neither published elsewhere nor submitted for publication simultaneously. None of the authors has any potential financial conflict of interest related to this manuscript.
Review
1- The authors already submitted this manuscript and received the proper comments from the reviewers including mines. Unfortunately, zero effort was made by this team to improve the quality and soundness of the scientific side of their investigation and fix its substantial flaws. The authors should consider in the future rigorously respond to the comments they receive accordingly.
Answer: First and foremost, we sincerely appreciate the invaluable contributions provided. Regrettably, we were unable to identify specific previous inquiries corresponding to the reviewer's, and thus, we do not have precise knowledge of the particular queries referenced in this response. Nonetheless, acknowledging that the primary concern raised in prior reviews was the lack of clarity regarding our objective in studying each HPV oncoprotein individually and in isolation, we believe that the current version addresses and elucidates the intent of our study. With this objective in mind, the utilization of HPV virus-bearing cell lines (as requested in previous reviews) could not be employed as it would yield a collective, rather than individual, response of the oncoproteins in the immune response. We apologize for not meeting your expectations with the revised manuscript. We acknowledge the feedback provided and regret that our efforts did not adequately address the concerns raised. Thank you for your feedback, and we will strive to improve in future submissions.
The complete contact information for corresponding author is: Dr. Antonio Carlos de Freitas (ORCID0000-0002-4957-9549), Av. Prof. Moraes Rego, 1235, Cidade Universitária, 50670-901, Recife-PE, Brazil. Fax: +55 81 21268512. E-mail: [email protected] .
Please do not hesitate to contact me if further information is needed.
Sincerely,
ANTONIO CARLOS DE FREITAS, PH.D
Associate Professor
Head of Laboratory of Molecular Studies and Experimental Therapy (LEMTE)
Department of Genetics
Federal University of Pernambuco
Recife, Pernambuco -Brazil

Reviewer 3 Report (New Reviewer)
Comments and Suggestions for Authors
I thank the Editor for giving me the oportunity to review the manuscript entitled "Immunological response against breast lineage cells transfected with Human Papillomavirus (HPV)" and the authors for their work.
HPV DNA has been been reported to be present in breast cancer samples, but a causal association, if any, is still under investigation and remains controversial. Santos et al. evaluated the effect on peripheral blood mononuclear cells of HPV E5, E6, and E7 oncogenes transfected in a cell line of breast cancer. In general the manuscript is well written, but I have some comments:
1- HPV is missing among the keywords.
2- Lines 43-44, "In 2020, 2.3 million women were diagnosed with breast cancer, with a mortality rate of 685,000 worldwide [2]": reference [2] does not seem to contain that data. Can the authors provide the precise reference they used for? Is it available online?
3- Line 45, please add "Brazilian" to "National Cancer Institute".
4- Lines 45-46 "in 2022, Brazil had an incidence of 73,610 cases of this type of cancer, and 17,825 of associated deaths [3]": article in [3] refers to year 2020.
5- Lines 41-46, please review the usage of terms "incidence" and "mortality rate". They are not used properly in the manuscript.
6- Line 56, "hr-HPVs": please define the abbreviation.
7- Line 87, "10% fetal bovine serum": why did authors decide to use 10% instead of 5% FBS for MDA-MB-231? Wasn't that concentration enough?
8- Line 136, "Cq values": the abbreviation has not been defined.
9- Lines 144-145, "the immune cells were collected and marked for immunophenotyping": did the authors manage to collect only immune cells or they collected a mix of the different cultured cells?
10- Lines 167-168, "30000 events in the P1 region (lymphocytes or monocytes gates) for cellular investigation and 2100 event": why did the authors choose those event rates?
11- Figure 2: please uniform the usage of Ct and Cq terms.
12- Figure 2: the figure shows a different number of replicates for each target (E5, E6, E7), why?
13- Line 250 is not readable.
14- Lines 378-379, "the results demonstrated that the main role of HPV in MDA-MB-231 cells is immunosuppression": I would rephrase it since authors did not evaluate all the possible effects in MDA-MB-231 cells induced by HPV.
15- I suggest authors to tone down the parts of the manuscript regarding any correlation between HPV and breast lesions. As stated in the abstract, this relationship remains controversial.
16- Did the authors identify limitations in the present study?
17- A data availability statement is missing in the manuscript.
Comments on the Quality of English Language
Minor English language editing is required.
Author Response
REPLY TO REFEREES
Recife, April 15, 2024
I am re-submitting the review article entitled “IMMUNOLOGICAL RESPONSE AGAINST BREAST LINEAGE CELLS TRANSFECTED WITH HUMAN PAPILLOMAVIRUS (HPV)” by Santos and São Marcos et al., for publication in Viruses.
All modifications are highlighted in the manuscript.
All the authors confirm that they saw and agreed to the submitted paper. The authors have been recognized as contributors and have agreed to their inclusion. The material is original, and it has been neither published elsewhere nor submitted for publication simultaneously. None of the authors has any potential financial conflict of interest related to this manuscript.
Review
- HPV is missing among the keywords.
Answer: We've added HPV to the keywords.
- Lines 43-44, "In 2020, 2.3 million women were diagnosed with breast cancer, with a mortality rate of 685,000 worldwide [2]": reference [2] does not seem to contain that data. Can the authors provide the precise reference they used for? Is it available online?
Answer: The precise reference has been added, as requested.
- Line 45, please add "Brazilian" to "National Cancer Institute".
Answer: The term has been added.
- Lines 45-46 "in 2022, Brazil had an incidence of 73,610 cases of this type of cancer, and 17,825 of associated deaths [3]": article in [3] refers to year 2020.
Answer: We have updated the reference in question.
- Lines 41-46, please review the usage of terms "incidence" and "mortality rate". They are not used properly in the manuscript.
Answer: The excerpt and terms questioned have been corrected.
6- Line 56, "hr-HPVs": please define the abbreviation.
Answer: The abbreviation has been defined as suggested.
7- Line 87, "10% fetal bovine serum": why did authors decide to use 10% instead of 5% FBS for MDA-MB-231? Wasn't that concentration enough?
Answer: The MDA-MB-231 cells were adapted to DMEM medium supplemented with 10% fetal bovine serum, as described in a study published by Passos, C.L.A., Polinati, R.M et al (2023). Additionally, in a study by Seung Wook Kim et al. (2015), it was demonstrated that the 10% fetal bovine serum concentration has a pronounced effect on the gene expression of MDA-MB-231 cancer cells.
Passos, C.L.A., Polinati, R.M., Ferreira, C. et al. Curcumin and melphalan cotreatment induces cell cycle arrest and apoptosis in MDA-MB-231 breast cancer cells. Sci Rep 13, 13446 (2023). https://doi.org/10.1038/s41598-023-40535-5.
Seung wook kim, Sun-jin Kim, Robert R. Langley and Isaiah J. Fidler. Modulation of the câncer cell transcriptome by culture media formulations and cell density. 46 (5): 2067-2075 (2015). https:// doi: 10.3892/ijo.2015.2930
8- Line 136, "Cq values": the abbreviation has not been defined.
Answer: The term "Cq" has been replaced with the abbreviation "Ct," and its definition has been added.
9- Lines 144-145, "the immune cells were collected and marked for immunophenotyping": did the authors manage to collect only immune cells or they collected a mix of the different cultured cells?
Answer: The highlighted statement has been corrected to eliminate any doubts about the procedure
10- Lines 167-168, "30000 events in the P1 region (lymphocytes or monocytes gates) for cellular investigation and 2100 event": why did the authors choose those event rates?
Answer: In our group, we acquire flow cytometry data in two ways: "All events," where we capture a large volume of events, typically between 500 thousand and 1 million, or specific gates for each cell population. Given that size and granularity characteristics allow us to distinguish between different populations, such as lymphocytes and monocytes, our preference is for targeted acquisition, using specific gates for each of them. Considering that the number of monocytes is generally lower compared to lymphocytes, our standard practice is to acquire 30 thousand events within the gate of each population to ensure a representative and accurate reading.
11- Figure 2: please uniform the usage of Ct and Cq terms.
Answer: We have corrected the terms as requested
12- Figure 2: the figure shows a different number of replicates for each target (E5, E6, E7), why?
Answer: Figure 2 illustrates the confirmation of oncogene transfection in MDA-MB-231 cells, which was conducted independently from the co-culture groups.
13- Line 250 is not readable.
Answer: We have made adjustments to the text to enhance its clarity.
14- Lines 378-379, "the results demonstrated that the main role of HPV in MDA-MB-231 cells is immunosuppression": I would rephrase it since authors did not evaluate all the possible effects in MDA-MB-231 cells induced by HPV.
Answer: We have considered rephrasing the statement to reflect the specific findings of the study accurately, considering that not all possible effects induced by HPV in MDA-MB-231 cells were evaluated.
15- I suggest authors to tone down the parts of the manuscript regarding any correlation between HPV and breast lesions. As stated in the abstract, this relationship remains controversial.
Answer: The suggestion has been implemented. We have toned down the sections of the manuscript concerning the correlation between HPV and breast lesions, considering the controversial nature of this relationship.
16- Did the authors identify limitations in the present study?
Answer: A major limitation of the study lies in assessing the tumor microenvironment of breast cancer using monolayer cell line cultures. Despite efforts to mimic the tumor microenvironment in vitro, we acknowledge that there are several other factors beyond the cancer cells and immune cells used in this study that need to be considered. However, the aim of this work was to evaluate whether the E5, E6, and E7 proteins of HPV, known for their oncogenic and immunosuppressive properties in well-established cancers, could modulate the response of lymphocytes and monocytes co-cultured with a breast cancer cell line. In this regard, the study proved to be effective, yielding results that can guide future research for a better understanding of the correlation between HPV and breast tumors.
17- A data availability statement is missing in the manuscript.
Answer: We did not include this item as it does not adhere to the format of the journal. However, we will provide the data upon request as per our standard protocol for such situations.
The complete contact information for corresponding author is: Dr. Antonio Carlos de Freitas (ORCID0000-0002-4957-9549), Av. Prof. Moraes Rego, 1235, Cidade Universitária, 50670-901, Recife-PE, Brazil. Fax: +55 81 21268512. E-mail: [email protected] .
Please do not hesitate to contact me if further information is needed.
Sincerely,
ANTONIO CARLOS DE FREITAS, PH.D
Associate Professor
Head of Laboratory of Molecular Studies and Experimental Therapy (LEMTE)
Department of Genetics
Federal University of Pernambuco
Recife, Pernambuco -Brazil

This manuscript is a resubmission of an earlier submission. The following is a list of the peer review reports and author responses from that submission.
Round 1
Reviewer 1 Report
Comments and Suggestions for Authors
Based on the association between HPV infection and breast cancer development discussed in the literature, the authors investigated the influence of HPV16 oncogene expression on the immune profile of PBMCs co-cultured with the breast cancer cell line MDA-MB-231. In summary, they observed that viral oncogene positive cancer cells are able to downregulate several immune cell functions. There are several major concerns regarding the study design.
1) Use of MDA-MB-231: When HPV oncogenes are expressed in a cancer cell line, there are some effects on each cancer cell line that can be measured. Therefore, the observations in MDA-MB-231 cells are not necessarily tissue-specific. However, further studies are needed to demonstrate tissue specificity and to link oncogenic functions with breast cancer. This requires immunostaining of HPV16-positive and HPV-negative tumor material for the significant observation made here. If this material is not available, co-cultures with HPV16 oncogene-expressing primary breast cancer cells must be performed. The results of these experiments must then be compared and discussed with the data previously described for primary oropharyngeal and primary cervical cells.
2) The Figures were not labeled correctly. Figure 4 D and E are not described in the text.
I therefore cannot support publication of this manuscript in its current form.
Comments on the Quality of English Language
The structure of the manuscript, specifically regarding English writing, requires strong revision.
Author Response
Reviewer 1
1) Use of MDA-MB-231: When HPV oncogenes are expressed in a cancer cell line, there are some effects on each cancer cell line that can be measured. Therefore, the observations in MDA-MB-231 cells are not necessarily tissue-specific. However, further studies are needed to demonstrate tissue specificity and to link oncogenic functions with breast cancer. This requires immunostaining of HPV16-positive and HPV-negative tumor material for the significant observation made here. If this material is not available, co-cultures with HPV16 oncogene-expressing primary breast cancer cells must be performed. The results of these experiments must then be compared and discussed with the data previously described for primary oropharyngeal and primary cervical cells.
Answer: We are grateful for the suggestions presented, and although we agree that with these suggestions our study woold be enriched. We understand that the results as presented are relevant enough to be published. The relationship between HPV and breast cancer is still not well-established. Studies show significant variability in the percentage of virus presence in breast tissues. Therefore, clarification of this relationship is of interest to the scientific community. In our study, we started with the idea that, before studying live tissues, using the cellular model to guide our research would be beneficial. The MDA-MB-231 cell line is a triple-negative breast cancer cell line widely used for such studies, and here, it was employed to understand the modulation of the tumor microenvironment (TME) caused by HPV in this type of breast cancer. In our work, we were able to demonstrate that HPV modulated the TME and may worsen the prognosis for patients with triple-negative breast cancer. Armed with this information, subsequent in vivo studies can be conducted to further contribute to clarifying the relationship between HPV and breast cancer. The techniques suggested for tissue studies represent greater effort and will benefit from the results obtained in the initial study here presented.
2) The Figures were not labeled correctly. Figure 4 D and E are not described in the text.
Answer: We included the call for figures in the text
The structure of the manuscript, specifically regarding English writing, requires strong revision.
Answer: We reviewed the English in the entire paper as requested.
The complete contact information for corresponding author is: Dr. Antonio Carlos de Freitas (ORCID0000-0002-4957-9549), Av. Prof. Moraes Rego, 1235, Cidade Universitária, 50670-901, Recife-PE, Brazil. Fax: +55 81 21268512. E-mail: [email protected] .
Please do not hesitate to contact me if further information is needed.
Sincerely,
ANTONIO CARLOS DE FREITAS, PH.D
Associate Professor
Head of Laboratory of Molecular Studies and Experimental Therapy (LEMTE)
Department of Genetics
Federal University of Pernambuco
Recife, Pernambuco -Brazil

Reviewer 2 Report
Comments and Suggestions for Authors
Breast cancer is the most common tumor worldwide. In the multifactorial etiology of this cancer, oncogenic viruses, especially HPVs, are also taken into account. TME is a complex ecosystem surrounding a tumor, composed of a variety of non-cancerous cells including blood vessels, immune cells, fibroblasts, signaling molecules and the extracellular matrix. As many studies show, HPV causes remodeling of the TME. Cell lines have been long used in order to study various tumors.
The Authors evaluated the role of HPV in breast adenocarcinoma cell line (MDA-MB-231), transfected with the HPV16 E5, E6, and E7 oncogenes and co-cultured with PBMCs. Immunophenotyping was performed.
They observed that transfected cells, especially E6 and E7, showed an increase of the suppressor profile of CD4 lymphocytes.
Additionally, the following cytokines were tested in the cell culture supernatant: IL-2, IL-4, IL-6, IL-10, TNF-α, IFN-α and IL-17.Authors concluded, MDA-MB-231 breast cell line transfected with HPV oncogenes can downregulate the number and function both lymphocytes and monocytes.
Main comments:
1. Figure 8. Cytokines are produced in the supernatants of lymphocyte and tumor cell cultures. I think the point is that cytokines were measured in the culture supernatants of lymphocytes and cancer cells.
2. Figure 9 Cytokines are produced in the supernatants of monocyte and tumor cell cultures. Similar inaccuracy. Cytokines are produced by monocytes and cancer cells and were determined in the supernatant. These titles should be clarified.
Minor comments:
1. Line 289 - na presença de E6 e E7, should be corrected
2. Errors in references, e.g. items 12, 14, 26 - please check all references are correct
Author Response
Reviewer 2
- Figure 8. Cytokines are produced in the supernatants of lymphocyte and tumor cell cultures. I think the point is that cytokines were measured in the culture supernatants of lymphocytes and cancer cells.
Answer: We have corrected the caption as requested
- Figure 9 Cytokines are produced in the supernatants of monocyte and tumor cell cultures. Similar inaccuracy. Cytokines are produced by monocytes and cancer cells and were determined in the supernatant. These titles should be clarified.
Answer: We have corrected the caption as requested
- Line 289 - na presença de E6 e E7, should be corrected
Answer: We correct the language of the text
- Errors in references, e.g. items 12, 14, 26 - please check all references are correct
Anwser: We correct the requested references
The complete contact information for corresponding author is: Dr. Antonio Carlos de Freitas (ORCID0000-0002-4957-9549), Av. Prof. Moraes Rego, 1235, Cidade Universitária, 50670-901, Recife-PE, Brazil. Fax: +55 81 21268512. E-mail: [email protected] .
Please do not hesitate to contact me if further information is needed.
Sincerely,
ANTONIO CARLOS DE FREITAS, PH.D
Associate Professor
Head of Laboratory of Molecular Studies and Experimental Therapy (LEMTE)
Department of Genetics
Federal University of Pernambuco
Recife, Pernambuco -Brazil

Reviewer 3 Report
Comments and Suggestions for Authors
The manuscript by Santos et al. reports the effect of HPV-16 E6, E7 and E5 proteins on the highly metastatic breast cancer cell line MDA-MB-231 by analyzing the immune response dynamic of co-cultured PBMCs ex vivo. The data reported by the authors are interesting however the model they are trying to establish linking high-risk HPV infection to breast cancer development using their current cell system is confusing. Based on multiple previous peer-reviewed reports, HPV-16 E6, E7 and E5 are established as oncogenes that affect the immunosuppressive status of HPV-16 infected cells therefore this work could be reproduced ex vivo using any cell type including primary ones. In my opinion, this investigation should have been developed using actual HPV-16 infected cancer cells (i.e. SiHa, Ca Ski or else) which are isolated from HPV-16 driven tumors (but again, this work is somehow already done and published). Finally, the only rational way to link breast cancers to HPV infection(s) is to demonstrate irrevocably the HPV infection status of a breast cancer cell, any other way could simply be due to “shed” HPV oncogene(s) or Head/Neck (less likely genital) metastatic cells that relocate to the breast of the infected individual.
Specific concerns:
1. Reference 8 in the manuscript does not mention HPV infection of breast cells therefore the authors should be accurate. To date, no such infection is reported irrevocably in the literature.
2. It would have been more rational to confirm E6, E7 and E5 expression after transfection using protein detection methods (i.e. western blotting and/or fluorescence microscopy). mRNA production does not mean the proteins are efficiently expressed.
3. The conclusion section should be re-written as the data by Santos et al. does not demonstrate any relation between HPV and breast cancer however it confirm the immunosuppression by HPV-16 E6, E7 and E5 proteins that is reported previously.
4. It would be substantial for the manuscript to show actual raw data from flow cytometry assessments instead of just histograms after signals quantification.
Comments on the Quality of English LanguageEnglish usage is fine.
Author Response
Recife, January 31, 2024.
Viruses
REPLY TO REFEREES
Reviewer 3 - Round 1
1) The manuscript by Santos et al. reports the effect of HPV-16 E6, E7 and E5 proteins on the highly metastatic breast cancer cell line MDA-MB-231 by analyzing the immune response dynamic of co-cultured PBMCs ex vivo. The data reported by the authors are interesting however the model they are trying to establish linking high-risk HPV infection to breast cancer development using their current cell system is confusing. Based on multiple previous peer-reviewed reports, HPV-16 E6, E7 and E5 are established as oncogenes that affect the immunosuppressive status of HPV-16 infected cells therefore this work could be reproduced ex vivo using any cell type including primary ones. In my opinion, this investigation should have been developed using actual HPV-16 infected cancer cells (i.e. SiHa, Ca Ski or else) which are isolated from HPV-16 driven tumors (but again, this work is somehow already done and published). Finally, the only rational way to link breast cancers to HPV infection(s) is to demonstrate irrevocably the HPV infection status of a breast cancer cell, any other way could simply be due to “shed” HPV oncogene(s) or Head/Neck (less likely genital) metastatic cells that relocate to the breast of the infected individual.
Answer: We are grateful for the suggestions presented, and although we agree that with these suggestions our study woold be enriched. The relationship between HPV and breast cancer is still not well-established. Studies show significant variability in the percentage of virus presence in breast tissues. Therefore, clarification of this relationship is of interest to the scientific community. In our study, we started with the idea that, before studying live tissues, using the cellular model to guide our research would be beneficial. The MDA-MB-231 cell line is a triple-negative breast cancer cell line widely used for such studies, and here, it was employed to understand the modulation of the tumor microenvironment (TME) caused by HPV in this type of breast cancer. In this study, we aimed to observe the influence of the E5, E6, and E7 oncoproteins of the virus on the immune response of tumor cells from MDA-MB-231 breast cancer. The suggested cell model was not utilized, as it did not allow for the individual analysis of each oncoprotein but rather the combined activity of all three. Thus, we believe that the proposed model was suitable to assist in satisfactorily answering the question of this study: "What is the individual influence of the E5, E6, and E7 oncogenes on the modulation of the immune response in the tumor microenvironment of HPV-positive breast cells?". Armed with this information, subsequent in vivo studies can be conducted to further contribute to clarifying the relationship between HPV and breast cancer.
2) Reference 8 in the manuscript does not mention HPV infection of breast cells therefore the authors should be accurate. To date, no such infection is reported irrevocably in the literature.
Answer: We corrected the reference and the requested information.
3) It would have been more rational to confirm E6, E7 and E5 expression after transfection using protein detection methods (i.e. western blotting and/or fluorescence microscopy). mRNA production does not mean the proteins are efficiently expressed.
Answer: We appreciate the suggestion, and its implementation is of great value to confirm protein-level expression. However, we believe that the RT-qPCR technique is sufficient to confirm transfection, as demonstrated in the following studies: Gao et al. (2021); Hu et al. (2020) and Su-Hyeong et al. (2009) . Additionally, the pcDNA vector is a strong expression vector commonly used in transfection studies, as Wang et al. (2017). The reported studies are described below:
Gao Z-Y, Gu N-J, Wu M-Z, et al. Human papillomavirus16 E6 but not E7 upregulates GLUT1 expression in lung cancer cells by upregulating thioredoxin expression. Technology in Cancer Research & Treatment. 2021;20. doi:10.1177/15330338211067111
Hu Y, Wu MZ, Gu NJ, Xu HT, Li QC, Wu GP. Human papillomavirus 16 (HPV 16) E6 but not E7 inhibits the antitumor activity of LKB1 in lung cancer cells by downregulating the expression of KIF7. Thorac Cancer. 2020 Nov;11(11):3175-3180. doi: 10.1111/1759-7714.13640. Epub 2020 Sep 18. PMID: 32945133; PMCID: PMC7606012.
Su-Hyeong Kim, Jung-Min Oh, Jae-Hong No, Yung-Jue Bang, Yong-Sung Juhnn, Yong-Sang Song, Involvement of NF-κB and AP-1 in COX-2 upregulation by human papillomavirus 16 E5 oncoprotein, Carcinogenesis, Volume 30, Issue 5, May 2009, Pages 753–757, https://doi.org/10.1093/carcin/bgp066
Wang, X., Zhang, Z., Cao, H., Niu, W., Li, M., Xi, X. E., & Wang, J. (2017). Human papillomavirus type 16 E6 oncoprotein promotes proliferation and invasion of non‐small cell lung cancer cells through Toll‐like receptor 3 signaling pathway. Journal of Medical Virology, 89(10), 1852-1860. doi: https://doi.org/10.1002/jmv.24845
4) The conclusion section should be re-written as the data by Santos et al. does not demonstrate any relation between HPV and breast cancer however it confirm the immunosuppression by HPV-16 E6, E7 and E5 proteins that is reported previously.
Answer: We modified the conclusion of the manuscript in accordance with the suggested guidelines.
5) It would be substantial for the manuscript to show actual raw data from flow cytometry assessments instead of just histograms after signals quantification.
Answer: We added the requested information as supplementary material to the manuscript.
The complete contact information for corresponding author is: Dr. Antonio Carlos de Freitas (ORCID0000-0002-4957-9549), Av. Prof. Moraes Rego, 1235, Cidade Universitária, 50670-901, Recife-PE, Brazil. Fax: +55 81 21268512. E-mail: [email protected] .
Please do not hesitate to contact me if further information is needed.
Sincerely,
ANTONIO CARLOS DE FREITAS, PH.D
Associate Professor
Head of Laboratory of Molecular Studies and Experimental Therapy (LEMTE)
Department of Genetics
Federal University of Pernambuco
Recife, Pernambuco -Brazil

Round 2
Reviewer 1 Report
Comments and Suggestions for Authors
Unfortunately, I have to stick to my first opinion. Since no further experiments on the relevance of the results in vivo were added, I am in a position to reject the paper.
Author Response
Recife, January 31, 2024.
Viruses
REPLY TO REFEREES – ROUND 2
I am re-submitting the review article entitled “IMMUNOLOGICAL RESPONSE AGAINST BREAST LINEAGE CELLS TRANSFECTED WITH HUMAN PAPILLOMAVIRUS (HPV)” by Santos and São Marcos et al., for publication in Viruses.
All modifications are highlighted in the manuscript.
All the authors confirm that they saw and agreed to the submitted paper. The authors have been recognized as contributors and have agreed to their inclusion. The material is original, and it has been neither published elsewhere nor submitted for publication simultaneously. None of the authors has any potential financial conflict of interest related to this manuscript.
Reviewer 1 - Round 2
1) Unfortunately, I have to stick to my first opinion. Since no further experiments on the relevance of the results in vivo were added, I am in a position to reject the paper.
Answer: We appreciate your review and the suggested techniques. The discussion was of great significance and will certainly be incorporated into our future studies. However, for the proposed article, we believe it presents results of significant relevance regarding the influence of HPV oncoproteins in modulating the tumor microenvironment of breast cancer. This was achieved through the use of an in vitro model employing the MB-MDA-231 cell line transfected with HPV16 oncoproteins.
The complete contact information for corresponding author is: Dr. Antonio Carlos de Freitas (ORCID0000-0002-4957-9549), Av. Prof. Moraes Rego, 1235, Cidade Universitária, 50670-901, Recife-PE, Brazil. Fax: +55 81 21268512. E-mail: [email protected] .
Please do not hesitate to contact me if further information is needed.
Sincerely,
ANTONIO CARLOS DE FREITAS, PH.D
Associate Professor
Head of Laboratory of Molecular Studies and Experimental Therapy (LEMTE)
Department of Genetics
Federal University of Pernambuco
Recife, Pernambuco -Brazil
